# Alcohol consumption and the risk of gastric intestinal metaplasia in a U.S. Veterans population

Hudson M. Holmes[1], Andre G. Jove[1], Mimi C. Tan[2], Hashem B. El-Serag[2,3], Aaron P. Thrift[4,5]*

**1** Department of Medicine, Baylor College of Medicine, Houston, Texas, United States of America, **2** Section of Gastroenterology and Hepatology, Department of Medicine, Baylor College of Medicine, Houston, Texas, United States of America, **3** Houston VA HSR&D Center for Innovations in Quality, Effectiveness and Safety, Michael E. DeBakey Veterans Affairs Medical Center, Houston, Texas, United States of America, **4** Section of Epidemiology and Population Sciences, Department of Medicine, Baylor College of Medicine, Houston, Texas, United States of America, **5** Dan L Duncan Comprehensive Cancer Center, Baylor College of Medicine, Houston, Texas, United States of America

* aaron.thrift@bcm.edu

**Data Availability Statement:** There are legal restrictions on sharing data publicly. The dataset used for this study contains sensitive patient information that is potentially identifying. The Department of Veterans Affairs (VA) restricts

## Abstract

### Background

Chronic alcohol use is a risk factor for non-cardia gastric adenocarcinoma. However, it is less well understood whether alcohol use is a risk factor for premalignant mucosal changes, namely gastric intestinal metaplasia. We examined the association between various parameters of alcohol use and risk of gastric intestinal metaplasia.

### Methods

We used data from 2084 participants (including 403 with gastric intestinal metaplasia) recruited between February 2008-August 2013 into a cross-sectional study at the Michael E. DeBakey Veterans Affairs Medical Center in Houston, Texas. All participants underwent a study upper endoscopy with systematic gastric mapping biopsies. Cases had intestinal metaplasia on any non-cardia gastric biopsy. Participants self-reported lifetime history of alcohol consumption, along with other lifestyle risk factors, through a study survey. We calculated odds ratios (OR) and 95% confidence intervals (95% CI) for categories of average alcohol consumption using multivariable logistic regression, and restricted cubic spline regression to explore the potential shape of a dose-response relationship.

### Results

Compared to lifelong non-drinkers, individuals who consumed on average ≥28 drinks per week had no elevated risk for gastric intestinal metaplasia (adjusted OR, 1.27; 95% CI, 0.74–2.19). Based on a spline regression curve and its 95% CI, there was also no demonstrable association between cumulative lifetime alcohol consumption and risk of gastric intestinal metaplasia. Similarly, we found no association between beverage type (beer, wine, liquor/spirits) and risk for gastric intestinal metaplasia.

access to such data unless researchers meet specific criteria per VHA Directive 1605.01, Privacy and Release of Information, section 13e which includes a written request for the data, IRB approval with waiver for HIPAA authorization prior to the request for individually identifiable information, and approval by the VA. These national sharing policies and standards also apply to deidentified data. Data requests may be sent to: Center for Innovations In Quality, Effectiveness, and Safety (IQuESt) 2450 Holcombe Blvd Suite 01Y Houston, TX 77021 713-791-1414 Attn: Laura Petersen, MD, MPH MEDVAMC Associate Chief of Staff, Research; Director IQuESt. Laura. Petersen@VA.gov.

**Funding:** This work was supported in part by funding from the National Institute of Diabetes and Digestive and Kidney Diseases (DK056338) and the U.S. Department of Veterans Affairs (CIN 13-413).

**Competing interests:** The authors report no competing interests for this publication.

## Conclusions

Neither amount of alcohol consumed nor specific beverage type was associated with risk of gastric intestinal metaplasia.

## Introduction

Gastric adenocarcinoma (gastric cancer) is among the leading causes of cancer-related deaths worldwide [1]. Patients diagnosed with gastric cancer have a poor prognosis, especially in advanced stages, with an observed 5 year survival rate of 5–20% [2]. Gastric cancer pathogenesis involves a well-defined pathway from chronic gastritis leading to gastric intestinal metaplasia, which leads to gastric intraepithelial neoplasia (i.e., dysplasia), and ultimately gastric cancer [1–3]. Gastric intestinal metaplasia, the precursor to gastric cancer, is detected on gastric biopsies as mucus secreting goblet cells in the gastric mucosa [4].

Chronic *Helicobacter pylori* infection has long been established to cause inflammation of the gastric mucosa and subsequently, the development of gastric intestinal metaplasia [4–6]. In populations with high incidence of gastric cancer, there have been multiple studies that investigate environmental risk factors synergistic to *H. pylori* infections in regard to the development of gastric intestinal metaplasia [6, 7]. However, there are few studies that look primarily at these modifiable factors independent of *H. pylori* infection, and even fewer in western populations, where *H. pylori* infection rates are lower, that focus on lifestyle risk factors, such as alcohol use, and their effects on gastric intestinal metaplasia and gastric cancer independent of or synergistically with *H. pylori* infection.

Alcohol is classified as a class 1 carcinogen by the International Agency for Research on Cancer [8], and heavy alcohol consumption has been demonstrated to independently increase the risk of some but not all subtypes of gastric cancer [9]. A meta-analysis of 44 case-control and 15 cohort studies published through 2010 reported a modest positive relationship between "heavy" alcohol consumption (defined as ≥4 drinks/day) and the risk of non-cardia gastric adenocarcinoma when controlling for other lifestyle factors such as smoking and fruit and vegetable consumption (vs. non-drinkers; relative risk, 1.20; 95% confidence interval [CI], 1.01–1.44) [10]. Conversely, there seems to be no association between alcohol consumption and gastric cardia cancer [7, 11, 12]. Despite the link between heavy alcohol consumption and the development of non-cardia gastric adenocarcinoma, it is unclear whether this association is attributed to the development of gastric intestinal metaplasia, or the progression of gastric intestinal metaplasia to gastric cancer. Few studies have examined the association between alcohol consumption and gastric intestinal metaplasia, with conflicting data. Additionally, those studies have been limited; either to non-U.S. populations or by the absence of data on a dose-dependent relationship between alcohol consumption and gastric intestinal metaplasia. Few studies have examined the relationship with specific beverage type [13–15].

We examined the relationship between alcohol consumption and gastric intestinal metaplasia, and specifically sought to quantify whether increased overall alcohol consumption or consumption of specific beverage types predispose to an increased risk of non-cardia gastric intestinal metaplasia in U.S populations. Because categorization may obscure potentially important differences in risk within and across categories of alcohol consumption, we also modeled the risk of non-cardia gastric intestinal metaplasia associated with alcohol consumption as a continuous measure and examined for linear or nonlinear patterns of risk associated with higher levels of alcohol consumption.

## Methods

### Study population

We used data from a cross-sectional study conducted among Veterans at the Michael E. DeBakey Veterans Affairs Medical Center (MEDVAMC) in Houston, Texas from February 2008-August 2013 [16, 17]. We recruited participants from two sources: (i) among consecutive eligible patients undergoing an elective esophagogastroduodenoscopy (EGD) for any indication; and (ii) among consecutive patients attending one of seven selected primary care clinics. These two patient groups represent the source population for non-cardia gastric intestinal metaplasia cases at the MEDVAMC. The patients recruited from primary care clinics were eligible for a routine (i.e., average risk) screening colonoscopy and we invited them to participate in the study, which required them to undertake an EGD for the study. None of the primary care patients were primarily referred for EGD and none were approached during a time of a pre-scheduled colonoscopy. Patients recruited from primary care underwent the study EGD and routine screening colonoscopy in the same endoscopic session. Criteria for eligibility included: (1) age 50–80 years (40–80 years for the elective EGD group); (2) no previous gastro-esophageal surgery; (3) no previous gastroesophageal cancer; (4) no active lung, liver, colon, breast or stomach cancer; (5) no anticoagulants; (6) no significant liver disease indicated by platelet count below 70 000 ascites, or known gastroesophageal varices; and (7) no history of major stroke or mental condition. The study was approved by the Institutional Review Boards for MEDVAMC and Baylor College of Medicine (H-27828). All participants provided written informed consent to take part in the study. Overall, 70% of patients in the elective EGD group and 43% of eligible patients in the primary care group underwent the study EGD and completed the study survey.

### Study procedure

During the study EGD, at least 10 biopsies (2 biopsies at each of the 5–7 biopsy sites according to adoption of the updated Sydney System [18]) were taken from the antrum (both greater and lesser curvature), corpus (proximal greater curvature, proximal lesser curvature, with optional additional biopsies at distal greater curvature and distal lesser curvature), and cardia. We systematically recorded endoscopic findings from the EGD. Biopsy specimens were embedded in paraffin, oriented on edge, sectioned in 5-sections, and stained with hematoxylin and eosin, alcian blue at pH 2.5. A modified silver stain and alcian blue–periodic acid Schiff stain were also used when staining for *H. pylori* was negative. Two gastrointestinal pathologists independently determined presence and severity of gastric intestinal metaplasia on each specimen. When necessary a third gastrointestinal pathologist was consulted to make the final decision. All pathologists were blinded to endoscopic findings and patient survey responses. Participants with evidence of intestinal metaplasia on ≥1 non-cardia gastric biopsy were classified as gastric intestinal metaplasia cases. Cases were compared to controls, defined as participants without intestinal metaplasia on all non-cardia gastric biopsies.

Patients were considered to have *H. pylori* infection if *H. pylori* organisms were isolated on gastric tissue culture or found on histopathology of ≥1 gastric biopsy site (using hematoxylin and eosin, alcian blue at pH 2.5, a modified silver stain, or alcian blue–periodic acid Schiff stain). To process cultures for *H. pylori*, frozen tissue specimens were thawed, homogenized, and inoculated onto Brain Heart Infusion medium (nutrient rich agar ideal for culturing fastidious microorganisms) and *H. pylori* Special Peptone Agar plates with 7% horse blood. The plates were incubated at 37˚C under micro-aerophilic conditions (5% $O_2$, 10% $CO_2$, and 85% $N_2$) in an Anoxomat jar for up to two weeks. Positive growth was transferred to a fresh,

nonselective Brain Heart Infusion blood agar plate and incubated for 48–72 hours. *H. pylori* were identified when the oxidase, catalase, and urease reactions were positive with a compatible Gram stain. To obtain a pure culture, we selected and subcultured several small round colonies from each patient's plate 1 or 2 times. Isolated strains were then stored at 80˚C in cysteine storage medium containing 20% glycerol.

## Data collection

The survey was completed prior to the study EGD with assistance from trained research staff. The survey ascertained information about age, sex, race/ethnicity, education, use of alcohol and smoking, medical history, and use of medications. We calculated body mass index (BMI) from pre-study EGD height and weight measurements. Waist and hip circumferences were measured via flexible tape measure and we calculated waist-to-hip ratio (WHR) as waist circumference divided by hip circumference. A ratio of $\geq 0.9$ and $\geq 0.85$ was used to categorize a high WHR for males and females, respectively.

For alcohol consumption, participants reported whether they currently drank alcohol, were life-long non-drinkers, or had previously drunk alcohol but stopped. Ever drinkers then reported if they had ever consumed alcohol at least monthly for $\geq 6$ months. Participants with a history of monthly alcohol consumption for $\geq 6$ months subsequently reported frequency of consumption for four classes of alcohol (beer, white wine, red wine and liquor/spirits) at ages 20–29, 30–49 and $\geq 50$ years, as applicable. Alcohol consumption (measured as number of all standard drinks [10 g alcohol/drink] summed across all classes of alcohol for all age groups) was divided by duration of drinking (since 20 years of age, in weeks) to calculate total lifetime alcohol consumption. Similar algorithms were used to calculate average lifetime beverage-specific consumptions [19]. In the analysis, participants who reported ever consuming alcohol but who did not consume alcohol at least monthly for $\geq 6$ months were considered non-drinkers and included in the referent group.

## Statistical analysis

Our primary aim was to estimate the relative risk of non-cardia gastric intestinal metaplasia associated with average lifetime alcohol consumption, total and beverage specific. We used unconditional multivariate logistic regression to calculate odds ratios (ORs) and associated 95% CIs. Total alcohol consumption was first modeled as a five-level categorical variable (life-long non-drinkers; $<7$; $7\text{-}<14$; $14\text{-}<28$, and $\geq 28$ drinks/week). We used the same 5 categories for beer consumption, but we used fewer categories for wine (life-long non-drinkers; $<7$; $\geq 7$) and liquor/spirits (life-long non-drinkers; $<7$; $7\text{-}<21$; $\geq 21$) due to limited range for these alcohol types. For all analyses, we compared history of alcohol consumption among ever drinkers (current and ex-drinkers) to a referent group of those who were lifelong non-drinkers. The final multivariate models were adjusted for potential confounders, including age (years), sex, race/ethnicity, highest level of education, BMI, smoking status, and *H. pylori* infection. To test for trend, category of alcohol consumption was included in the multivariate model as an ordinal variable (with category values taking the median of the range observed among the control group) and non-drinkers were excluded. We included participants with missing data for covariates (e.g., BMI, WHR, *H. pylori* infection, smoking status) in the analyses using an additional category for missing values.

Furthermore, to explore the shape of the dose-response association between alcohol consumption and risk of gastric intestinal metaplasia, we used a logistic regression model with restricted cubic spline for average lifetime alcohol consumption (in drinks/week) as a continuous measure by means of generalized additive logistic models (using the CRAN package mgcv

in R software), adjusted for the same covariates. Smoothing splines fixed at 3 degrees of freedom were used to test for significance of nonlinearity against the linear effect [20–22].

Finally, we assessed potential biological interaction between alcohol consumption and (i) smoking status and (ii) *H. pylori* infection status in relation to the risk of non-cardia gastric intestinal metaplasia by creating new variables that reclassified participants according to their combined exposure and tested for departure from additivity using the Synergy Index (SI). A SI greater than 1 indicates that the joint effect of two risk factors on the risk of gastric intestinal metaplasia is greater than the sum of their independent effects and suggests the presence of biological interaction [23].

All analyses were conducted using Stata 13.0 (StataCorp LP, College Station, TX) and all tests for statistical significance two-sided at $\alpha = 0.05$.

## Results

This study included data from 2084 participants, with 1568 recruited from EGD clinics and 516 from primary care. The mean age of participants was 60.2 years (standard deviation, 8.1 years). Ninety-two percent of participants were male, 57.3% White and 31.3% Black. Most participants were overweight or obese (81.4%) and reported a history of smoking (68.5%).

Among the 2084 included participants, 403 were classified as cases with gastric intestinal metaplasia and 1681 as controls without gastric intestinal metaplasia. The demographic characteristics of cases and controls are shown in Table 1. Compared to controls, cases with non-cardia gastric intestinal metaplasia were significantly more likely to be aged ≥60 years at study enrollment and male, but less likely to be White and have a college degree (Table 1). Cases were also more likely to have *H. pylori* infection (52.5% vs. 21.9%) and a history of cigarette smoking (80.7% vs. 71.0%).

Overall, 91.5% of study participants reported a history of alcohol consumption. Controls were slightly more likely than cases with non-cardia gastric intestinal metaplasia to report being a lifelong non-drinker (9.0% vs. 6.4%). Among controls who reported being ever drinkers, the median lifetime total alcohol consumption was 13 (interquartile range [IQR], 6–42) and 4 (IQR, 2–10) drinks/week for males and females, respectively. Median lifetime total alcohol consumption was numerically higher among male (16 drinks/week; IQR, 6–42) and female (8 drinks/week; IQR, 2–14) cases with non-cardia gastric intestinal metaplasia than controls.

Table 2 shows the unadjusted and adjusted ORs for associations between alcohol consumption and risk of gastric intestinal metaplasia. We found no association between alcohol drinking status and risk of non-cardia gastric intestinal metaplasia (current drinkers vs. lifelong non-drinkers; adjusted OR, 1.03; 95% CI, 0.63–1.68). Likewise, there was no association with average amount of alcohol consumed over the life-course, even for heavy drinkers. Compared to lifelong non-drinkers, individuals who consumed on average ≥28 drinks per week had no elevated risk for non-cardia gastric intestinal metaplasia (adjusted OR, 1.27; 95% CI, 0.74–2.19). Among alcohol drinkers, the risk for gastric intestinal metaplasia did not increase linearly with increasing total average alcohol consumption (p-trend = 0.24). When we analyzed beverage-specific alcohol consumption, there were no associations of beer, wine or liquor consumption with the risk of non-cardia gastric intestinal metaplasia (Table 2).

We also examined the risk of non-cardia gastric intestinal metaplasia associated with average lifetime total alcohol consumption as a continuous measure using both linear and nonlinear regression functions and assigning lifelong non-drinkers a value 0 drinks/week. Fig 1 shows the fitted spline regression curve and its 95% CI, indicating no association with gastric intestinal metaplasia across the full range of average consumption (up to an average of 140 drinks/week). There was no evidence for a nonlinear dose relationship (p = 0.46 testing for departure from linearity).

**Table 1. Characteristics of cases and controls.**

| | | Controls | Cases | |
|---|---|---|---|---|
| | | N = 1,681 | N = 403 | |
| | | N (%) | N (%) | Pᵃ |
| Age group, years | | | | 0.001 |
| | <60 | 715 (42.5) | 134 (33.3) | |
| | 60–69 | 787 (46.8) | 206 (51.1) | |
| | ≥70 | 179 (10.7) | 63 (15.6) | |
| Sex | | | | <0.001 |
| | Male | 1524 (90.7) | 391 (97.0) | |
| | Female | 157 (9.3) | 12 (3.0) | |
| Race/Ethnicity | | | | <0.001 |
| | White | 1028 (61.2) | 165 (40.9) | |
| | Black | 484 (28.8) | 169 (41.9) | |
| | Other | 169 (10.0) | 69 (17.1) | |
| Highest level of education | | | | <0.001 |
| | Less than high school | 94 (5.9) | 46 (12.0) | |
| | High school graduate | 575 (36.3) | 178 (46.2) | |
| | Tech/vocational college | 601 (37.9) | 104 (27.0) | |
| | College graduate | 314 (19.8) | 57 (14.8) | |
| | Missing | 97 | 18 | |
| BMI, kg/m² | | | | 0.06 |
| | <25 | 300 (17.9) | 87 (21.6) | |
| | 25–29.9 | 595 (35.5) | 153 (38.0) | |
| | ≥30 | 783 (46.7) | 163 (40.4) | |
| | Missing | 3 | 0 | |
| WHR | | | | 0.94 |
| | Low | 241 (14.6) | 58 (14.8) | |
| | High | 1404 (85.4) | 334 (85.2) | |
| | Missing | 36 | 11 | |
| *H. pylori* infection | | | | <0.001 |
| | No | 1295 (78.1) | 189 (47.5) | |
| | Yes | 363 (21.9) | 209 (52.5) | |
| | Missing | 23 | 5 | |
| Smoking status | | | | <0.001 |
| | Never smoker | 456 (29.0) | 74 (19.3) | |
| | Ex-smoker | 638 (40.5) | 166 (43.2) | |
| | Current smoker | 480 (30.5) | 144 (37.5) | |
| | Missing | 107 | 19 | |
| Alcohol drinking status | | | | 0.19 |
| | Non-drinker | 151 (9.0) | 26 (6.4) | |
| | Ex-drinker | 639 (38.0) | 166 (41.2) | |
| | Current drinker | 891 (53.0) | 211 (52.4) | |

BMI, body mass index; WHR, waist-to-hip ratio.

WHR was categorized as high if it was ≥0.9 for males or ≥0.85 for females.

ᵃ P-values from chi-square tests for categorical variables; missing category was excluded from statistical tests for differences between controls and cases.

**Table 2. Unadjusted and adjusted odds ratios for associations between alcohol-related variables and risk of gastric intestinal metaplasia.**

| | | Controls | Cases | Unadjusted | Adjusted |
|---|---|---|---|---|---|
| | | N | N | OR (95% CI) | OR[a] (95% CI) |
| Current alcohol drinking status | | | | | |
| | Non-drinker[b] | 151 | 26 | 1.00 (ref) | 1.00 (ref) |
| | Ex-drinker | 639 | 166 | 1.51 (0.96–2.37) | 1.13 (0.69–1.85) |
| | Current drinker | 891 | 211 | 1.38 (0.88–2.14) | 1.03 (0.63–1.68) |
| Average lifetime total alcohol consumption (drinks/week) | | | | | |
| | <7 | 432 | 94 | 1.30 (0.80–2.11) | 1.00 (0.59–1.70) |
| | 7 to <14 | 222 | 58 | 1.56 (0.93–2.62) | 1.22 (0.69–2.17) |
| | 14 to <28 | 196 | 56 | 1.70 (1.01–2.88) | 1.16 (0.65–2.07) |
| | ≥28 | 394 | 119 | 1.80 (1.12–2.90) | 1.27 (0.74–2.19) |
| | *p*-trend[c] | | | 0.05 | 0.24 |
| Average lifetime beer consumption (drinks/week) | | | | | |
| | <7 | 483 | 123 | 1.52 (0.94–2.44) | 1.12 (0.66–1.89) |
| | 7 to <14 | 140 | 47 | 2.00 (1.16–3.45) | 1.44 (0.78–2.64) |
| | 14 to <28 | 151 | 43 | 1.70 (0.98–2.94) | 1.19 (0.65–2.19) |
| | ≥28 | 311 | 84 | 1.61 (0.98–2.64) | 1.11 (0.63–1.96) |
| | *p*-trend[c] | | | 0.93 | 0.79 |
| Average lifetime wine consumption (drinks/week) | | | | | |
| | <7 | 174 | 54 | 1.85 (1.09–3.14) | 1.47 (0.77–2.82) |
| | ≥7 | 87 | 24 | 1.64 (0.88–3.07) | 1.19 (0.55–2.54) |
| | *p*-trend[c] | | | 0.67 | 0.42 |
| Average lifetime liquor consumption (drinks/week) | | | | | |
| | <7 | 406 | 102 | 1.50 (0.92–2.43) | 1.08 (0.62–1.87) |
| | 7 to <21 | 185 | 45 | 1.45 (0.84–2.49) | 0.95 (0.51–1.77) |
| | ≥21 | 179 | 58 | 1.93 (1.14–3.26) | 1.37 (0.75–2.50) |
| | *p*-trend[c] | | | 0.15 | 0.18 |

CI, confidence interval; OR, odds ratio.

[a] Adjusted for age, sex, race/ethnicity, highest level of education, body mass index, smoking status, and *H. pylori* infection.

[b] Reference group for all analyses is non-drinkers.

[c] *p* for trend excludes non-drinkers.

As cases were more likely than controls to have a history of cigarette smoking and *H. pylori* infection, we examined for potential biological interactions between smoking and alcohol consumption and *H. pylori* and alcohol consumption. The highest risk for gastric intestinal metaplasia was observed among ever smokers with a history of alcohol consumption of ≥28 drinks/week (Table 3). Compared to individuals who were lifelong non-drinkers and also never smoked, individuals who were ever smokers and consumed on average ≥28 drinks/week of alcohol had 2-fold higher risk for gastric intestinal metaplasia (OR, 1.89; 95% CI, 1.00–3.20). However, when formally tested using the SI, we found no evidence for biological interaction of smoking with alcohol suggesting the majority of the increased risk is conferred by smoking, not alcohol (SI, 0.88; 95% CI, 0.13–6.06). Likewise, we found no evidence for biological interaction of *H. pylori* infection with alcohol consumption on risk of gastric intestinal

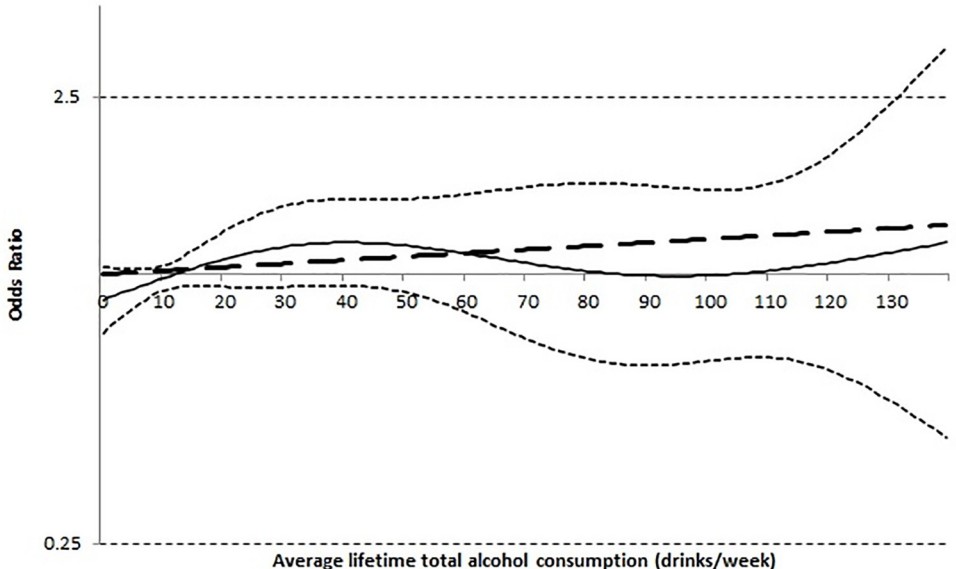

**Fig 1. Dose-response relation of average lifetime total alcohol consumption (drinks/week) on risk of gastric intestinal metaplasia.** (a) solid line = odds ratio; (b) dotted line = 95% confidence interval for odds ratio; (c) dashed line = null hypothesis of no association; (d) extended dashed line = linear association.

metaplasia (SI, 1.41; 95% CI 0.52–3.79). *H. pylori* infection conferred higher risk of gastric intestinal metaplasia irrespective of amount of alcohol consumed (Table 4).

## Discussion

In this large cross-sectional study, we examined whether a relationship existed between quantity and type of alcohol consumption and the risk of non-cardia gastric intestinal metaplasia. While there was tentative evidence for a synergistic effect between tobacco smoke and total alcohol consumption on the risk of gastric intestinal metaplasia, total alcohol consumption alone was not associated with increased risk for gastric intestinal metaplasia. Most of the excess

**Table 3. Association of combined alcohol and smoking exposure and risk of gastric intestinal metaplasia.**

| | Controls | Cases | | |
|---|---|---|---|---|
| | N | N | Unadjusted | Adjusted |
| | | | OR (95% CI) | OR[a] (95% CI) |
| Never smoker, non-drinker | 91 | 14 | 1.00 (ref) | 1.00 (ref) |
| Never smoker, <14 drinks/week | 174 | 29 | 1.08 (0.55–2.15) | 1.02 (0.50–2.11) |
| Never smoker, 14 to <28 drinks/week | 40 | 5 | 0.81 (0.27–2.41) | 0.61 (0.20–1.90) |
| Never smoker, ≥28 drinks/week | 53 | 14 | 1.72 (0.76–3.88) | 1.49 (0.63–3.55) |
| Ever smoker, non-drinker | 52 | 10 | 1.25 (0.52–3.01) | 1.28 (0.50–3.25) |
| Ever smoker, <14 drinks/week | 474 | 121 | 1.66 (0.91–3.01) | 1.41 (0.75–2.65) |
| Ever smoker, 14 to <28 drinks/week | 154 | 50 | 2.11 (1.11–4.03) | 1.68 (0.85–3.34) |
| Ever smoker, ≥28 drinks/week | 336 | 105 | 2.03 (1.11–3.72) | 1.89 (1.00–3.20) |
| SI (95% CI) | | | | 0.88 (0.13–6.06) |

CI, confidence interval; OR, odds ratio; SI, synergy index.

[a] Adjusted for age, sex, race/ethnicity, highest level of education, body mass index, and *H. pylori* infection.

**Table 4. Association of combined alcohol and *H. pylori* infection exposure and risk of gastric intestinal metaplasia.**

| | Controls | Cases | Unadjusted | Adjusted |
|---|---|---|---|---|
| | N | N | OR (95% CI) | OR[a] (95% CI) |
| *H. pylori* negative, non-drinker | 115 | 11 | 1.00 (ref) | 1.00 (ref) |
| *H. pylori* negative, <14 drinks/week | 503 | 72 | 1.50 (0.77–2.91) | 1.17 (0.58–2.33) |
| *H. pylori* negative, 14 to <28 drinks/week | 143 | 24 | 1.75 (0.82–3.73) | 1.31 (0.60–2.86) |
| *H. pylori* negative, ≥28 drinks/week | 304 | 50 | 1.72 (0.86–3.42) | 1.22 (0.59–2.51) |
| *H. pylori* positive, non-drinker | 28 | 13 | 4.85 (1.97–12.0) | 4.04 (1.59–10.2) |
| *H. pylori* positive, <14 drinks/week | 140 | 78 | 5.82 (2.96–11.5) | 3.94 (1.94–8.01) |
| *H. pylori* positive, 14 to <28 drinks/week | 51 | 30 | 6.15 (2.86–13.2) | 3.81 (1.71–8.51) |
| *H. pylori* positive, ≥28 drinks/week | 84 | 68 | 8.46 (4.22–17.0) | 5.59 (2.67–11.7) |
| SI (95% CI) | | | | 1.41 (0.52–3.79) |

CI, confidence interval; OR, odds ratio; SI, synergy index.

[a] Adjusted for age, sex, race/ethnicity, highest level of education, body mass index, and smoking status.

risk among individuals who smoked and consumed alcohol was the result of smoking, not alcohol consumption. Likewise, when stratified by specific beverage type, we found no associations between consumption of beer, wine and liquor/sprits and gastric intestinal metaplasia.

The role of alcohol in the pathway to development of gastric adenocarcinoma is still unclear and discrepant amongst particular cancer sub-types [2, 10, 12, 24]. Two meta-analyses conducted by Tramacere et al. concluded that "heavy" alcohol consumption (≥4 drinks per day) was associated with increased risk of non-cardia gastric adenocarcinoma [10] whereas there appears to be no association between any degree of alcohol consumption and the risks of cardia gastric adenocarcinoma or esophageal adenocarcinoma [12]. This discrepancy supports a protocol in which epidemiologic studies evaluating risk factors for gastric cancer specify regionality of the stomach being investigated. Additionally, this framework should be applied when considering risk factors for premalignant conditions, such as gastric intestinal metaplasia. Thus, our study specifically explored alcohol's relationship, if any, with development of non-cardia gastric intestinal metaplasia.

Alcohol is a known carcinogen [8, 25, 26] and has been associated with many acute pathologic changes in the gut including, but not limited to, hemorrhagic lesions of the gastric mucosa [27] and dose-dependent modulatory effects on gastric acid secretion [28–32]. Given this, and alcohol's association with non-cardia gastric adenocarcinoma previously mentioned, it is reasonable to inquire whether alcohol is also a risk factor for precursor lesions, such as intestinal metaplasia. Previous literature was sparse, especially in western populations, but predominately indicated no association between alcohol consumption and metaplasia formation [13, 14]. These prior studies alone are insufficient, however, in that neither regarded alcohol consumption as the primary variable of interest and neither stratified alcohol usage based off quantity consumed or beverage type. Given the understanding that effects of alcohol on the gastrointestinal tract are largely dependent on a variety of factors such as dose and lifetime use [33], further studies are warranted to adequately examine for any potential relationship. Our study used numerous stratification schemes to sub-classify alcohol use and search for an association with morphologic changes of the gastric mucosa. No degree of alcohol use correlated to any increased risk for non-cardia gastric intestinal metaplasia development, which provides the strongest evidence to date that there is no independent association between the two.

With our results promoting the absence of an association between alcohol and non-cardia gastric intestinal metaplasia and prior evidence reporting a relationship between heavy alcohol

consumption and non-cardia gastric cancer [10], it is plausible to hypothesize that alcohol contributes to the later stages of the non-cardia gastric adenocarcinoma development pathway. Previous studies of upper gastrointestinal tract cancers found that alcohol is a strong risk factor for development of esophageal squamous cell carcinoma [26] and late premalignant conditions such as high grade dysplasia. However, there was no association with early precursor lesions such as low grade dysplasia of the esophagus [34]. This model does offer an interesting framework for how alcohol could plausibly pose a risk in later stages of gastric carcinogenesis. However, further studies are necessary to conclude the presence or absence of any such relationship.

This study has a number of strengths, such as well-defined cases and controls, specific classifications for categorical variables used in the multivariate analysis, and a large sample size. These factors contributed to this study's ability to fully investigate any potential relationship between varying quantities of alcohol consumption, or consumption of any specific alcoholic beverage and the risk of gastric intestinal metaplasia. The comprehensive pre-study questionnaire provided to subjects allowed for extensive control of potential confounders. This provides assurance that data ascertained from this study directly represents the interplay between alcohol and non-cardia gastric intestinal metaplasia risk. Further, the administration of this questionnaire prior to the study EGD served to limit recall bias in the responses [19], which could otherwise alter the validity of the data. Finally, all EGD procedures and histological classifications of samples were based on established protocols [18]; this coupled with the blinding of the pathologists prevents potential sources of misclassification bias.

Limitations of this study include generalizability of the study population and potential heterogeneity of gastric intestinal metaplasia cases. The VA population predominately consisted of white males, which could hinder this study's ability to make statements regarding this relationship in other ethnicities or in women. The overall response rate among the primary care group was 43% and this may have biased our results. However, participants who consent are generally healthier than the general population and, if one existed, this would strengthen an association. In regard to positive cases, the parameter set to define a case was a finding of metaplasia on ≥1 non-cardia gastric biopsy. While this aided our study in searching for any preliminary relationship, further studies could look more specifically at extensiveness of metaplasia, specific locations of metaplasia, and subtype of metaplasia. We included in our referent group a fraction of participants who may have ever consumed alcohol but who did not consume alcohol at least monthly for ≥6 months during their lifetime. As a result, we may have biased our risk estimates towards the null. Additionally, the risk estimates may be subject to reporting bias due to self-report alcohol intake. Although we attempted to limit this bias by having participants complete the study surveys prior to the study EGD (and knowledge of case-control study), and the bias in exposure status would likely be non-differential between cases and controls, it may have biased the association toward the null. Diet is an important risk factor for gastric cancer; however, we did not include diet in our study. Nonetheless, lack of adjustment for diet would not explain the null association because, as a confounder, diet would instead have exaggerated an observed association, instead of biasing it toward the null. Finally, as with all cross-sectional investigations, this study cannot demonstrate a causal relationship between two variables–only a relative strength or lack of association. A natural consequence of the cross-sectional design also meant that we could not match controls with cases. Nonetheless, this should not undermine the importance of the observed lack of association.

## Conclusions

In conclusion, no specific beverage type or quantity of alcohol consumption posed any increased risk for non-cardia gastric intestinal metaplasia. Further, there was no demonstrable

risk between total lifetime alcohol consumption and the same outcome. While there was some evidence of an additive effect between tobacco smoke and alcohol consumption on non-cardia gastric intestinal metaplasia risk, alcohol use to any degree alone may not confer increased risk for gastric intestinal metaplasia.

## Author Contributions

**Conceptualization:** Aaron P. Thrift.

**Data curation:** Mimi C. Tan, Aaron P. Thrift.

**Formal analysis:** Aaron P. Thrift.

**Funding acquisition:** Hashem B. El-Serag.

**Investigation:** Hudson M. Holmes, Andre G. Jove, Mimi C. Tan, Hashem B. El-Serag, Aaron P. Thrift.

**Methodology:** Hashem B. El-Serag, Aaron P. Thrift.

**Supervision:** Aaron P. Thrift.

**Writing – original draft:** Hudson M. Holmes, Andre G. Jove, Mimi C. Tan, Hashem B. El-Serag, Aaron P. Thrift.

**Writing – review & editing:** Hudson M. Holmes, Andre G. Jove, Mimi C. Tan, Hashem B. El-Serag, Aaron P. Thrift.

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
