## [Decision Letter · Decision Letter 0]

12 Jul 2021

PONE-D-21-15870

No association between alcohol consumption and the risk of gastric intestinal metaplasia

PLOS ONE

Dear Dr. Thrift,

Thank you for submitting your manuscript to PLOS ONE. After careful consideration, we feel that it has merit but does not fully meet PLOS ONE’s publication criteria as it currently stands. Therefore, we invite you to submit a revised version of the manuscript that addresses the points raised during the review process.

The manuscript has ben sent out to two outstanding external referees with a strong research background in epidemiology of alcohol consumption. I would invite the authors to carefully consider their comments attached below. In addition to the reviewers’ comments, this editor had a few points to be addressed:

Estimates were not adjusted for a proxy of socioeconomic status (SES), that is, the main determinants of gastric cancer. This was an important comment from Reviewer #1 that greatly limits your results. Please address this bias.For what regards the dose-risk relationships, the figure appears to depict a linear trend. In fact, when I go through the manuscript text, I found the author’s sentence “There was no evidence for a nonlinear dose relationship (p=0.46 […]”. Therefore, it does not make sense to report a nonlinear trend when there was no evidence against linearity. Therefore, in figure 1 please report a linear relationship and use a logarithm scale for the y-axis. In fact, in the current figure version, the y axis range is inappropriate and not on a log-scale.On page 13, the authors wrote “Among alcohol drinkers, the risk for gastric intestinal metaplasia did not increase linearly with increasing total average consumption (p-trend=0.14)”. There is something unclear as it seems that the trend cannot be considered non-linear being the p-value greater than 0.14. Please clarify.Among the limitations, I would state that controls were not matched to cases, as a “natural” consequence of the cross-sectional design.

We look forward to receiving your revised manuscript.

Kind regards,

Matteo Rota, Ph.D.

Academic Editor

PLOS ONE

Journal Requirements:

2.  Please include in your Methods section the date ranges over which you recruited participants to this study.

Furthermore, in your Methods section, please provide a justification for the sample size used in your study, including any relevant power calculations (if applicable).

Reviewers' comments:

Reviewer's Responses to Questions

**Comments to the Author**

1. Is the manuscript technically sound, and do the data support the conclusions?

Reviewer #1: Partly

Reviewer #2: Yes

2. Has the statistical analysis been performed appropriately and rigorously? 

Reviewer #1: Yes

Reviewer #2: Yes

3. Have the authors made all data underlying the findings in their manuscript fully available?

Reviewer #1: Yes

Reviewer #2: No

4. Is the manuscript presented in an intelligible fashion and written in standard English?

Reviewer #1: Yes

Reviewer #2: Yes

5. Review Comments to the Author

Reviewer #1: The authors found no association between cumulative lifetime alcohol intake and risk of gastric intestinal metaplasia.

Some remarks:

1) The authors stated that “participants who reported ever consuming alcohol but who did not consume alcohol at least monthly for ≥6 months were considered non-drinkers and included in the referent group”. An additional analysis in which these individuals are excluded from the referent group is appropriate. Please, make it available.

2) The authors stated that “The referent group for all analyses was those who were lifelong non-drinkers”. I suppose that ex-drinkers are not included in the referent group. This is correct, but I suggest to clary state this fact.

3) No measures/indexes of social status have been included as potential confounders. Due to the high importance that social status has as potential confounder for the association of alcohol intake with health outcome, authors should make any effort to include it in the multivariable model. On the contrary, the absence of any index of social status in the panel of covariates must be discussed as a strong limitation of the manuscript.

4) Another potentially important variable the authors should adjusted their models for is BMI, also because, beyond formal statistical significance, BMI shows some degree of association with case/control status (Table 1).

5) “Missing category was excluded from statistical tests for differences between controls and cases” (Table 1). However, it is not clear how the authors dealt with missing values in multivariable analyses (adjusted also for smoking and HP infection, that have missings). Please clarify. A case complete approach is not appropriate, and some more robust approach (multiple imputation, for example) for dealing with missing values have to be implemented.

6) The authors repeatedly have used terms as “(statistically) significant” or “non-significant”. It appears that some of their conclusions have been based only on statistically significance. This is not correct. Please, avoid it and base your conclusion more on effect size

7) Authors should provide complete data for biological interaction of H. pylori infection with alcohol consumption on risk of gastric intestinal metaplasia, in the same way they provided in Table 3 data on the association of combined alcohol and smoking exposure

Reviewer #2: This cross-sectional study investigated the relationship between self-reported lifetime alcohol consumption and beverage-specific consumption and gastric intestinal metaplasia in ~2000 US Veterans. This study is clearly written with a well-defined and justified research question (to look at alcohol’s role in precursor of gastric cancer rather than gastric cancer itself, which could add to understanding of alcohol’s role in gastric cancer development), and the statistical methods appeared to be sound. However, the study would benefit from further clarification of the study methods and importantly on the discussions of potential limitations of generalisability and self-reported bias of alcohol consumption. Please see the full list of suggestions below.

Title: I would suggest to avoid stating the conclusion (e.g. no association), but to include the study design and study population in the title.

Abstract:

1. It was slightly confusing to read the number of cases and number of controls separately in the first place, and later realise that it was a cross-sectional study but not a case-control study. I would suggest to first report the overall sample size recruited, and then report the number of cases identified.

2. Please state the full name of Houston VA Medical Center to make it clear the study population was among veterans.

3. Please state the study period (in Abstract and also in the Methods later).

Methods:

1. There was no mentioning of missing data until missing data was shown in Table 1. Please indicate in the Statistical analysis the number of participants with missing data and how missing data was handled in each of the subsequent analyses.

2. Page 7, please move the sentence “Overall, 70% of patients in the elective EGD …” up to the end of the Study population paragraph on page 6.

3. How many participants were recruited from each of the elective EGD group and of the primary care group, respectively?

4. Please specify how much one drink quantifies (e.g., in grams or units).

Results:

1. Page 11, please report the overall sample size and provide a summary of descriptive data of the whole study population in the beginning, before moving on to describe cases and controls separately.

2. Table 1 title: The terms case and controls are self-explanatory, so there is no need to repeat “controls without” and “cases with gastric intestinal metaplasia”. I would suggest to change the title along the lines of “Characteristics of cases and controls” or “Characteristics of US Veterans by case status”

3. Table 1: Please indicate the method used to derive the p-values.

4. Page 13, please report IQR for median of alcohol intake.

5. Table 2: Please be consistent with terminology e.g. non-drinkers (table and main text) vs. never drinkers (footnote).

6. Figure 1 title: Please remove “compared with controls” from the title.

Discussion

1. The authors briefly mentioned generalizability of the study population (predominantly white males) as a limitation. However, other aspect of the study population (Veterans only), as well as the different response rate between the elective EGD group and the primary care group (if the socio-demographics or health seeking behaviours differ between the two groups), may also have implications on the generalizability to the general population and bias. It would be good to see some discussion in the paper on these points.

2. Please discuss the potential bias of self-reported alcohol intake in the limitations.

3. Diet is an important risk factor for stomach cancer but was not measured in the study – this should be discussed in the limitation.

4. Page 20, final sentence “alcohol use to any degree alone proved to be an insufficient predisposing factor”. The word “proved” seems a bit too strongly conclusive, especially in the context of cross-sectional study - a more suggestive tone would be more appropriate.

6. PLOS authors have the option to publish the peer review history of their article (what does this mean?). If published, this will include your full peer review and any attached files.

Reviewer #1: **Yes: **Augusto Di Castelnuovo

Reviewer #2: No

---

## [Author Response · Author response to Decision Letter 0]

9 Sep 2021

July 21, 2021

Matteo Rota, Ph.D.

Academic Editor

PLOS ONE

Dear Dr. Rota,

Thank you for considering our manuscript, and for the helpful comments from the reviewers which accompanied your recent letter. We have taken the opportunity of revising our manuscript in accordance with those comments and herewith submit it to you for further consideration. 

Comments from the Editor

Comment #1:

Estimates were not adjusted for a proxy of socioeconomic status (SES), that is, the main determinants of gastric cancer. This was an important comment from Reviewer #1 that greatly limits your results. Please address this bias.

Reply to Comment #1:

We thank the Editor and Reviewer for this important comment. We captured educational background in our study survey, which we now use here as a proxy for social status. As expected, we show in Table 1 that highest level of education is strongly associated with case-control status (cases are more likely to have high school or less education). We have therefore added highest level of education as a covariate in our multivariable models. Additionally adjusting for education level did attenuate the associations with alcohol consumption to the null. We believe adding this data to the paper strengthens our methods and helps to strengthen confidence in our findings.

Comment #2:

For what regards the dose-risk relationships, the figure appears to depict a linear trend. In fact, when I go through the manuscript text, I found the author’s sentence “There was no evidence for a nonlinear dose relationship (p=0.46 […]”. Therefore, it does not make sense to report a nonlinear trend when there was no evidence against linearity. Therefore, in figure 1 please report a linear relationship and use a logarithm scale for the y-axis. In fact, in the current figure version, the y axis range is inappropriate and not on a log-scale.

Reply to Comment #2:

A common approach to the analysis of alcohol consumption in cancer (and other disease) studies has been to categorize participants based on broad measures of alcohol intake. Such an approach has the advantage of deriving an easily interpretable measure of relative risk, and makes no assumption about any underlying trends or patterns of risk. However, categorization may obscure potentially important differences in risk within and across categories of alcohol intake, particularly if the cut-off points for categorization span biologically important thresholds. Modeling risk with alcohol intake as a continuous measure by using standard linear regression models avoids the potential loss of information owing to categorization, but it constrains the dose effect to be linear. Again, such constraints may mask important differences in risk associated with different levels of alcohol intake. As has been well described, alcohol consumption has a J-shaped relationship with all-cause mortality in human beings and there is no reason to discount the possibility that similar nonlinear patterns of risk may be observed for stomach cancer and its precursor, gastric intestinal metaplasia. In the current manuscript, we use a suite of approaches to address this issue. In total, our analyses consistently demonstrate no association with alcohol consumption. The splines demonstrate no hidden/missing non-linear relationship. The lack of linear association across categories is accurate, and there is no ‘hidden’ patterns of risk obscured by the categorization or assumption of linearity.

Comment #3:

On page 13, the authors wrote “Among alcohol drinkers, the risk for gastric intestinal metaplasia did not increase linearly with increasing total average consumption (p-trend=0.14)”. There is something unclear as it seems that the trend cannot be considered non-linear being the p-value greater than 0.14. Please clarify.

Reply to Comment #3:

There is actually no association with alcohol consumption. The associations with categories of alcohol consumption demonstrated a lack of linear association (no increase with increasing category of exposure). The splines show that there is also no non-linear association. Together, these analyses address issues of categorization and assumptions of linearity. They strengthen our conclusion that alcohol consumption does not increase a person’s risk for gastric intestinal metaplasia.

Comment #4:

Among the limitations, I would state that controls were not matched to cases, as a “natural” consequence of the cross-sectional design.

Reply to Comment #4:

We added the following to the Discussion (version with tracked changes; page 24, paragraph 1):

“A natural consequence of the cross-sectional design also meant that we could not match controls with cases.”

Reviewer #1

Comment #1:

The authors stated that “participants who reported ever consuming alcohol but who did not consume alcohol at least monthly for ≥6 months were considered non-drinkers and included in the referent group”. An additional analysis in which these individuals are excluded from the referent group is appropriate. Please, make it available.

Reply to Comment #1:

Unfortunately, the alcohol questions in our study survey were written such that we did not capture any alcohol history among persons who may have ever consumed alcohol but who did not consume alcohol at least monthly for ≥6 months. This was the qualifying question for capturing specific alcohol consumption. We have now noted this as a limitation of our study (page 23, paragraph 2):

“We included in our referent group a fraction of participants who may have ever consumed alcohol but who did not consume alcohol at least monthly for ≥6 months during their lifetime. As a result, we may have biased our risk estimates towards the null.”

Comment #2:

The authors stated that “The referent group for all analyses was those who were lifelong non-drinkers”. I suppose that ex-drinkers are not included in the referent group. This is correct, but I suggest to clary state this fact.

Reply to Comment #2:

The Reviewer is correct. Ex-drinkers were not included in the referent group. We revised the Methods (page 10, paragraph 2) as follows:

“For all analyses, we compared history of alcohol consumption among ever drinkers (current and ex-drinkers) to a referent group of those who were lifelong non-drinkers.”

Comment #3:

No measures/indexes of social status have been included as potential confounders. Due to the high importance that social status has as potential confounder for the association of alcohol intake with health outcome, authors should make any effort to include it in the multivariable model. On the contrary, the absence of any index of social status in the panel of covariates must be discussed as a strong limitation of the manuscript.

Reply to Comment #3:

We thank the Reviewer for this important comment. We captured educational background in our study survey, which we now use here as a proxy for social status. As expected, we show in Table 1 that highest level of education is strongly associated with case-control status (cases are more likely to have high school or less education). We have therefore added highest level of education as a covariate in our multivariable models. Additionally adjusting for education level did attenuate the risk estimates for associations with alcohol consumption to the null.

Comment #4:

Another potentially important variable the authors should adjusted their models for is BMI, also because, beyond formal statistical significance, BMI shows some degree of association with case/control status (Table 1).

Reply to Comment #4:

We agree with the Reviewer and have now adjusted our models for BMI.

Comment #5:

“Missing category was excluded from statistical tests for differences between controls and cases” (Table 1). However, it is not clear how the authors dealt with missing values in multivariable analyses (adjusted also for smoking and HP infection, that have missings). Please clarify. A case complete approach is not appropriate, and some more robust approach (multiple imputation, for example) for dealing with missing values have to be implemented.

Reply to Comment #5:

We apologize for excluding this information from the Methods. As shown in Table 1, we had missing data on BMI, WHR, H. pylori infection, and smoking status among a minority of study participants. We added the following to the Methods (page 10, paragraph 2):

“We included participants with missing data for covariates (e.g., BMI, WHR, H. pylori infection, smoking status) in the analyses using an additional category for missing values.”

Comment #6:

The authors repeatedly have used terms as “(statistically) significant” or “non-significant”. It appears that some of their conclusions have been based only on statistically significance. This is not correct. Please, avoid it and base your conclusion more on effect size.

Reply to Comment #6:

We have revised our manuscript throughout to remove emphasis on “statistical significance”. 

Comment #7:

Authors should provide complete data for biological interaction of H. pylori infection with alcohol consumption on risk of gastric intestinal metaplasia, in the same way they provided in Table 3 data on the association of combined alcohol and smoking exposure.

Reply to Comment #7:

We have now added this data in Table 4 (page 20).

Reviewer #2

Title: I would suggest to avoid stating the conclusion (e.g. no association), but to include the study design and study population in the title.

Reply: We have changed our title to “Alcohol consumption and the risk of gastric intestinal metaplasia in a U.S. Veterans population”.

Abstract:

1. It was slightly confusing to read the number of cases and number of controls separately in the first place, and later realise that it was a cross-sectional study but not a case-control study. I would suggest to first report the overall sample size recruited, and then report the number of cases identified.

Reply: We revised the Abstract (page 3) as follows:

“We used data from 2084 participants (including 403 with gastric intestinal metaplasia) recruited between February 2008-August 2013 into a cross-sectional study at the Michael E. DeBakey Veterans Affairs Medical Center in Houston, Texas.”

2. Please state the full name of Houston VA Medical Center to make it clear the study population was among veterans.

Reply: We replaced “Houston VA Medical Center” with “Michael E. DeBakey Veterans Affairs Medical Center in Houston, Texas”.

3. Please state the study period (in Abstract and also in the Methods later).

Reply: We added the study period to both the Abstract (page 3) and Methods (page 7).

Methods:

1. There was no mentioning of missing data until missing data was shown in Table 1. Please indicate in the Statistical analysis the number of participants with missing data and how missing data was handled in each of the subsequent analyses.

Reply: We apologize for excluding this information from the Methods. As shown in Table 1, we had missing data on BMI, WHR, H. pylori infection, and smoking status among a minority of study participants. We added the following to the Methods (page 10, paragraph 2):

“We included participants with missing data for covariates (e.g., BMI, WHR, H. pylori infection, smoking status) in the analyses using an additional category for missing values.”

2. Page 7, please move the sentence “Overall, 70% of patients in the elective EGD …” up to the end of the Study population paragraph on page 6.

Reply: We moved this sentence as suggested.

3. How many participants were recruited from each of the elective EGD group and of the primary care group, respectively?

Reply: We added this information to the Results (page 12, paragraph 1):

“This study included data from 2084 participants, with 1568 recruited from EGD clinics and 516 from primary care.”

4. Please specify how much one drink quantifies (e.g., in grams or units).

Reply: We now noted in the Methods that one standard drink was 10g of alcohol.

Results:

1. Page 11, please report the overall sample size and provide a summary of descriptive data of the whole study population in the beginning, before moving on to describe cases and controls separately.

Reply: We revised the Results (page 12, paragraph 1) as follows:

“This study included data from 2084 participants, with 1568 recruited from EGD clinics and 516 from primary care. The mean age of participants was 60.2 years (standard deviation, 8.1 years). Ninety-two percent of participants were male, 57.3% White and 31.3% Black. Most participants were overweight or obese (81.4%) and reported a history of smoking (68.5%).”

2. Table 1 title: The terms case and controls are self-explanatory, so there is no need to repeat “controls without” and “cases with gastric intestinal metaplasia”. I would suggest to change the title along the lines of “Characteristics of cases and controls” or “Characteristics of US Veterans by case status”

Reply: We changed the title to “Characteristics of cases and controls.

3. Table 1: Please indicate the method used to derive the p-values.

Reply: We added this to the footnote of Table 1.

4. Page 13, please report IQR for median of alcohol intake.

Reply: We revised the results (page 15, paragraph 1) as follows:

“Among controls who reported being ever drinkers, the median lifetime total alcohol consumption was 13 (interquartile range [IQR], 6-42) and 4 (IQR, 2-10) drinks/week for males and females, respectively. Median lifetime total alcohol consumption was numerically higher among male (16 drinks/week; IQR, 6-42) and female (8 drinks/week; IQR, 2-14) cases with non-cardia gastric intestinal metaplasia than controls.”

5. Table 2: Please be consistent with terminology e.g. non-drinkers (table and main text) vs. never drinkers (footnote).

Reply: We placed “never drinkers” with “non-drinkers” in the footnote (page 17).

6. Figure 1 title: Please remove “compared with controls” from the title.

Reply: We removed “compared with controls” from the figure title.

Discussion

1. The authors briefly mentioned generalizability of the study population (predominantly white males) as a limitation. However, other aspect of the study population (Veterans only), as well as the different response rate between the elective EGD group and the primary care group (if the socio-demographics or health seeking behaviours differ between the two groups), may also have implications on the generalizability to the general population and bias. It would be good to see some discussion in the paper on these points.

Reply: We added the following to the Discussion (page 23, paragraph 2):

“The overall response rate among the primary care group was 43% and this may have biased our results. However, participants who consent are generally healthier than the general population and, if one existed, this would strengthen an association.”

2. Please discuss the potential bias of self-reported alcohol intake in the limitations.

Reply: We added the following to the Discussion (page 24, paragraph 1):

“Additionally, the risk estimates may be subject to reporting bias due to self-report alcohol intake. We attempted to limit this bias by having participants complete the study surveys prior to the study EGD (and knowledge of case-control study), increasing the likelihood that any bias would be non-differential between cases and controls.”

3. Diet is an important risk factor for stomach cancer but was not measured in the study – this should be discussed in the limitation.

Reply: We added the following to the Discussion (page 24, paragraph 1):

“Diet is an important risk factor for gastric cancer; however, we did not include diet in our study, which may have confounded our associations with alcohol consumption.”

4. Page 20, final sentence “alcohol use to any degree alone proved to be an insufficient predisposing factor”. The word “proved” seems a bit too strongly conclusive, especially in the context of cross-sectional study - a more suggestive tone would be more appropriate.

Reply: We revised the sentence (page 24, paragraph 2) as follows:

“While there was some evidence of an additive effect between tobacco smoke and alcohol consumption on non-cardia gastric intestinal metaplasia risk, alcohol use to any degree alone may not confer increased risk for gastric intestinal metaplasia.”

We look forward to your consideration of our revised manuscript, and trust that these revisions meet with your approval.

Yours sincerely,

Aaron P. Thrift, PhD

---

## [Decision Letter · Decision Letter 1]

7 Oct 2021

PONE-D-21-15870R1Alcohol consumption and the risk of gastric intestinal metaplasia in a U.S. Veterans populationPLOS ONE

Dear Dr. Thrift,

Thank you for submitting your manuscript to PLOS ONE. After careful consideration, we feel that it has merit but does not fully meet PLOS ONE’s publication criteria as it currently stands. Therefore, we invite you to submit a revised version of the manuscript that addresses the points raised during the review process.

There are still two minor comments that need to be better addressed before taking a positive decision on the manuscript. See below. This is a minor revision. In addition to these reviewer #2 comments, I would ask to the authors to better set the y-axis limits of Figure 1, now ranging from 0.1 to 10. I suggest to set a range from 0.25 to 4, using a logarithmic scale, as common for plots of dose-risk relationships.

I also want to point out that my previous comment #2 was partly misunderstood by the authors. The assessment of dose-risk relationship shape trough restricted cubic spline (rcs) showed no evidence of a non-linear association. This is fine, but from a statistical point of view, this means that there is no reason to consider a less parsimonious model (as the rcs is) when no evidence of a better fit emerged as compared to the linear line. However, I still agree with the author choice to show the rcs dose-risk relationship, but I also suggest to report the linear line in a supplementary figure, if not possible within Figure 1.

We look forward to receiving your revised manuscript.

Kind regards,

Matteo Rota, Ph.D.

Academic Editor

PLOS ONE

Journal Requirements:

Additional Editor Comments (if provided):

Reviewers' comments:

Reviewer's Responses to Questions

**Comments to the Author**

1. If the authors have adequately addressed your comments raised in a previous round of review and you feel that this manuscript is now acceptable for publication, you may indicate that here to bypass the “Comments to the Author” section, enter your conflict of interest statement in the “Confidential to Editor” section, and submit your "Accept" recommendation.

Reviewer #1: All comments have been addressed

Reviewer #2: (No Response)

2. Is the manuscript technically sound, and do the data support the conclusions?

Reviewer #1: Yes

Reviewer #2: Yes

3. Has the statistical analysis been performed appropriately and rigorously? 

Reviewer #1: Yes

Reviewer #2: Yes

4. Have the authors made all data underlying the findings in their manuscript fully available?

Reviewer #1: Yes

Reviewer #2: No

5. Is the manuscript presented in an intelligible fashion and written in standard English?

Reviewer #1: Yes

Reviewer #2: Yes

6. Review Comments to the Author

Reviewer #1: The authors provided a satisfactory revision

Reviewer #2: The authors have generally done well in revising the manuscript and responding to the reviewers’ comments. However, I still have a few minor comments:

1. Limitation paragraph, discussion on self-report bias in alcohol consumption: even if the bias in exposure status is non-differential between cases and controls, non-differential misclassification of the exposure can still bias the association toward the null.

2. Limitation paragraph, discussion on diet: The first half of the sentence “Diet is an important risk factor for gastric cancer; however, we did not include diet in our study…” sounds fine as poor diet/nutrition is associated with heavy drinking, and is a possible confounder in any observed association between alcohol consumption and gastric cancer-related outcome. However, the second part of the sentence “… which may have confounded our associations with alcohol consumption” is a bit be confusing in this context since there is no observed association between alcohol consumption and risk of gastric intestinal metaplasia in this study. I would suggest the authors to revise the second part of sentence to discuss this issue more specifically in the context of this study. For example, the lack of diet info in this study is unlikely to have major impact on your conclusions (since as a confounder diet would have exaggerated the observed associations, instead of biasing it toward the null).

7. PLOS authors have the option to publish the peer review history of their article (what does this mean?). If published, this will include your full peer review and any attached files.

Reviewer #1: **Yes: **Augusto Di Castelnuovo

Reviewer #2: No

---

## [Author Response · Author response to Decision Letter 1]

22 Oct 2021

October 20, 2021

Matteo Rota, Ph.D.

Academic Editor

PLOS ONE

Dear Dr. Rota,

Thank you for considering our revised manuscript, and for the additional comments in your recent letter. We have taken the opportunity of revising our manuscript in accordance with those comments and herewith submit it to you for further consideration. 

Comments from the Editor

Comment #1:

There are still two minor comments that need to be better addressed before taking a positive decision on the manuscript. See below. This is a minor revision. In addition to these reviewer #2 comments, I would ask to the authors to better set the y-axis limits of Figure 1, now ranging from 0.1 to 10. I suggest to set a range from 0.25 to 4, using a logarithmic scale, as common for plots of dose-risk relationships. 

I also want to point out that my previous comment #2 was partly misunderstood by the authors. The assessment of dose-risk relationship shape trough restricted cubic spline (rcs) showed no evidence of a non-linear association. This is fine, but from a statistical point of view, this means that there is no reason to consider a less parsimonious model (as the rcs is) when no evidence of a better fit emerged as compared to the linear line. However, I still agree with the author choice to show the rcs dose-risk relationship, but I also suggest to report the linear line in a supplementary figure, if not possible within Figure 1.

Reply to Comment #1:

We apologize for our misunderstanding your prior comment. We have revised the Figure to set the y-axis limits as suggested and added a linear line. We hope that this new Figure better reflects the findings from our study.

Reviewer #2

The authors have generally done well in revising the manuscript and responding to the reviewers’ comments. However, I still have a few minor comments:

Comment #1:

Limitation paragraph, discussion on self-report bias in alcohol consumption: even if the bias in exposure status is non-differential between cases and controls, non-differential misclassification of the exposure can still bias the association toward the null.

Reply to Comment #1:

Thank you for this important comment. We have revised the Discussion (page 22, paragraph 1) as follows:

“Additionally, the risk estimates may be subject to reporting bias due to self-report alcohol intake. Although we attempted to limit this bias by having participants complete the study surveys prior to the study EGD (and knowledge of case-control study), and the bias in exposure status would likely be non-differential between cases and controls, it may have biased the association toward the null.”

Comment #2:

Limitation paragraph, discussion on diet: The first half of the sentence “Diet is an important risk factor for gastric cancer; however, we did not include diet in our study…” sounds fine as poor diet/nutrition is associated with heavy drinking, and is a possible confounder in any observed association between alcohol consumption and gastric cancer-related outcome. However, the second part of the sentence “… which may have confounded our associations with alcohol consumption” is a bit be confusing in this context since there is no observed association between alcohol consumption and risk of gastric intestinal metaplasia in this study. I would suggest the authors to revise the second part of sentence to discuss this issue more specifically in the context of this study. For example, the lack of diet info in this study is unlikely to have major impact on your conclusions (since as a confounder diet would have exaggerated the observed associations, instead of biasing it toward the null).

Reply to Comment #2:

We have revised the sentence (page 22, paragraph 1) as follows:

“Diet is an important risk factor for gastric cancer; however, we did not include diet in our study. Nonetheless, lack of adjustment for diet would not explain the null association because, as a confounder, diet would instead have exaggerated an observed association, instead of biasing it toward the null.”

We look forward to your consideration of our revised manuscript, and trust that these revisions meet with your approval.

Yours sincerely,

Aaron P. Thrift, PhD

Associate Professor

Section of Epidemiology and Population Sciences

Department of Medicine

Baylor College of Medicine

---

## [Decision Letter · Decision Letter 2]

2 Nov 2021

Alcohol consumption and the risk of gastric intestinal metaplasia in a U.S. Veterans population

PONE-D-21-15870R2

Dear Dr. Thrift,

We’re pleased to inform you that your manuscript has been judged scientifically suitable for publication and will be formally accepted for publication once it meets all outstanding technical requirements.

Kind regards,

Matteo Rota, Ph.D.

Academic Editor

PLOS ONE

Additional Editor Comments (optional):

Reviewers' comments:

Reviewer's Responses to Questions

**Comments to the Author**

1. If the authors have adequately addressed your comments raised in a previous round of review and you feel that this manuscript is now acceptable for publication, you may indicate that here to bypass the “Comments to the Author” section, enter your conflict of interest statement in the “Confidential to Editor” section, and submit your "Accept" recommendation.

Reviewer #2: All comments have been addressed

2. Is the manuscript technically sound, and do the data support the conclusions?

Reviewer #2: (No Response)

3. Has the statistical analysis been performed appropriately and rigorously? 

Reviewer #2: (No Response)

4. Have the authors made all data underlying the findings in their manuscript fully available?

Reviewer #2: (No Response)

5. Is the manuscript presented in an intelligible fashion and written in standard English?

Reviewer #2: (No Response)

6. Review Comments to the Author

Reviewer #2: (No Response)

7. PLOS authors have the option to publish the peer review history of their article (what does this mean?). If published, this will include your full peer review and any attached files.

Reviewer #2: No

---

## [Editor Report · Acceptance letter]

5 Nov 2021

PONE-D-21-15870R2 

Alcohol consumption and the risk of gastric intestinal metaplasia in a U.S. Veterans population 

Dear Dr. Thrift:

I'm pleased to inform you that your manuscript has been deemed suitable for publication in PLOS ONE. Congratulations! Your manuscript is now with our production department. 

Kind regards, 

on behalf of

Dr. Matteo Rota 

Academic Editor

PLOS ONE